# Investigation of cell signalings and therapeutic targets in PTPRK-RSPO3 fusion-positive colorectal cancer

Jae Heon Jeong[ID][1,2,3☯], Jae Won Yun[ID][4☯], Ha Young Kim[2,3], Chan Yeong Heo[ID][3,5]*, Sejoon Lee[ID][6,7]*

**1** Integrated Major in Innovative Medical Science, College of Medicine, Seoul National University, Seoul, Republic of Korea, **2** Interdisciplinary Program for Bioengineering, College of Engineering, Seoul National University, Seoul, Republic of Korea, **3** Department of Plastic and Reconstructive Surgery, Seoul National University Bundang Hospital, Seongnam, Republic of Korea, **4** Veterans Medical Research Institute, Veterans Health Service Medical Center, Seoul, Korea, **5** Department of Plastic and Reconstructive Surgery, College of Medicine, Seoul National University, Seoul, Republic of Korea, **6** Precision Medicine Center, Seoul National University Bundang Hospital, Seongnam, South Korea, **7** Department of Pathology and Translational Medicine, Seoul National University Bundang Hospital, Seongnam, South Korea

☯ These authors contributed equally to this work.
* sejooning@gmail.com (SL); lionheo@gmail.com (CYH)

**Data Availability Statement:** All relevant data are within the paper and its Supporting Information files.

## Abstract

### Introduction

Colorectal cancer (CRC) is one of the most deadly and common diseases in the world, accounting for over 881,000 casualties in 2018. The *PTPRK-RSPO3* (P:R) fusion is a structural variation in CRC and well known for its ability to activate WNT signaling and tumorigenesis. However, till now, therapeutic targets and actionable drugs are limited in this subtype of cancer.

### Materials and method

The purpose of this study is to identify key genes and cancer-related pathways specific for P:R fusion-positive CRC. In addition, we also inferred the actionable drugs in bioinformatics analysis using the Cancer Genome Atlas (TCGA) data.

### Results

2,505 genes were altered in RNA expression specific for P:R fusion-positive CRC. By pathway analysis based on the altered genes, ten major cancer-related signaling pathways (Apoptosis, Direct p53, EGFR, ErbB, JAK-STAT, tyrosine kinases, Pathways in Cancer, SCF-KIT, VEGFR, and WNT-related Pathway) were significantly altered in P:R fusion-positive CRC. Among these pathways, the most altered cancer genes (*ALK*, *ACSL3*, *AXIN*, *MYC*, *TP53*, *GNAQ*, *ACVR2A*, and *FAS*) specific for P:R fusion and involved in multiple cancer pathways were considered to have a key role in P:R fusion-positive CRC. Based on the drug-target network analysis, crizotinib, alectinib, lorlatinib, brigatinib, ceritinib, erdafitinib, infigratinib and pemigatinib were selected as putative therapeutic candidates, since they

**Funding:** This study was supported by a VHS Medical Center Research Grant, Republic of Korea (VHSMC22057), grant no 18-2018-023 from the SNUBH Research Fund, and The National Research Foundation of Korea (NRF) grant funded by the Korea government (MSIT) (No. 2022R1C1C1012986). The funders had no role in study design, data collection and analysis, decision to publish, or preparation of the manuscript.

**Competing interests:** The authors have declared that no competing interests exist.

were already used in routine clinical practice in other cancer types and target genes of the drugs were involved in multiple cancer-pathways.

## Introduction

Colorectal cancer (CRC) is the third most fatal and fourth most diagnosed cancer worldwide, according to 2018 global cancer data released by the IARC. Approximately 2 million new cases were recorded in year 2018 alone, resulting in approximately 1 million fatalities [1–3]. With the development of NGS technology, simultaneous detection of various mutations in colorectal cancer has become possible, including SNV, INDEL, CNV, fusion, and MSI [4–7]. The reason that the detection of these mutations is important is that it can be used as a target therapy for gene mutations, for example, EGFR-inhibitor for *KRAS* wildtype CRC and immune checkpoint inhibitors for MSI-high solid tumor [8].

The efforts to develop targeted drugs for treating colorectal cancer are increasing, however, the candidates of target drugs other signaling pathways besides EGFR and mismatch Repair are limited. WNT signaling is known as a major pathway in colorectal cancer and mostly is activated by mutation of the *APC* gene, which plays an important role in the pathogenesis of colorectal cancer [9–13]. Recently, *PTPRK-RSPO3* (P:R) fusion also contributes to the activation of WNT signaling and causes colorectal cancer, and this mutation is mutually exclusive with the *APC* mutation and is recognized as another important mutation contributing to the development of colorectal cancer [14–17]. Recent studies have reported that LGK974 and *RSPO3* antibodies may be beneficial at in vitro and in vivo levels, however, the development of targeted therapeutics for colorectal cancer patients with P:R fusions is still in its infancy [18, 19].

Herein, we systematically inferred drug candidates in P:R fusion colorectal cancer. First, we extracted *RSPO3* expression correlated genes and selected oncogenic cell signal pathways containing those genes. Then, we constructed a drug-target network in P:R fusion colorectal cancer using the drug-target database, and finally, we prioritized a suitable therapeutic agent.

## Materials and methods

### Sample collection and quality control

The Broad GDAC Firehouse website (https://gdac.broadinstitute.org) provided gene level 3 (RSEM) mRNA expression with normalized read count values of the Cancer Genome Atlas (TCGA) colorectal cancer (CRC). The above-mentioned website provided information on the samples' MAF files, TNM stages, and molecular subtypes among other clinical characteristics.

### Case-control selection and selection of genes affected by PTPRK- RSPO3 fusion

We found seven samples with *PTPRK-RSPO3* fusion using the TCGA fusion gene data portal (The Jackson Laboratory, https://www.tumorfusions.org), which were cross-checked with elevated *RSPO3* expression levels. Of the 433 GDAC data downloaded, 53 normal and 1 metastasis data were omitted for the analysis of the remaining 379 data. For control sample selection, 50 samples were randomly selected among the samples with low *RSPO3* expression (less than the median value of *RSPO3* RNA expression, N = 186). To obtain R-values of 20,531 genes in correlation with *RSPO3* in RNA expression, Pearson correlation-tests were performed in seven *PTPRK-RSPO3* fusion-positive cases and 50 controls. Then, above tests were repeated 100

times. Based on the median of absolute R values from 100 tests, 20,531 genes were sorted in decreasing order. Using the median of absolute R values, mostly affected 2,505 genes were selected by correlation cut-off (R > 0.2). The R cut-off value, 0.2, were selected based on the 100,000 permutation tests. For each permutation test, randomly ordered expression values of a randomly selected gene were tested using Pearson correlation test with the expression values of reference gene (*RSPO3*) in 7 cases and 50 controls. After 100,000 tests, falsely selected genes correlated with *RSPO3* are assumed to be 0.37% when the cut-off value for R was 0.2.

### Pathway analysis via ConsensusPathDB (CPDB)

The aforementioned R-value data were used to perform over-representation analysis (ORA) using ConsensusPathDB (CPDB, http://cpdb.molgen.mpg.de/CPDB) on the 2,505 genes (including *RSPO3*). According to BioCarta [http://www.biocarta.com/], 177 biological pathways were combined from the following sources: INOH [20], KEGG [21], NetPath [22], PID [23], Reactome [24] and Wikipathways [25]. Analyzing the ontological features and the proportion of duplicated genes, the pathways enriched with chosen 2,505 genes (q-value < 0.05) were collapsed into 10 cancer-related pathways, having 848 genes as components.

### Inferring and prioritizing actionable drugs

The "Clinical Evidence Summaries" data was downloaded from the Clinical Interpretations of Variants in Cancer (CIViC) website (https://civic.genome.wustl.edu/releases) on July 1, 2021, and the "Actionable Variants" data was accessed and downloaded from the Precision Oncology Knowledge Base (OncoKB) website (http://oncokb.org/) on July 1, 2021. RSPO3-crrelated genes were annotated using 673 CIViC variations (181 genes) with predicted treatment effectiveness and 148 OncoKB actionable variants (53 genes). Then drug-target relationships were prioritized based on the scenario that properly working cancer drugs are generally inhibitors for activated oncogenes or activators for down-regulated tumor suppressor genes.

### Statistical analysis and data visualization

All statistical analyses, including the Pearson correlation-tests, were performed using the open software R version 3.4.4. Complexheatmap, a R package, was used to visualize an RNA expression heatmap. KEGG mapper (https://www.genome.jp/kegg/mapper.html) was used to display target genes associated to WNT signaling pathway. The comprehensive network between targetable drugs and therapeutic agents was analyzed and illustrated using Cytoscape 3.5.3. In this study, statistical significance was determined as a p-value of 0.05 and false detection rate (FDR) as a q-value of 0.2 in over-representation analysis.

## Results

### Clinico-pathological characteristics

The clinicopathological characteristics in this study were described in Table 1. A total 379 tumor solid samples, and of the 372 samples, excluding the 7 fusion-positive samples, 186 samples expressing less than 50% RSPO3 mRNA expression were selected as control.

There showed no definite statistical significance of histological type, age, sex, vital status and TNM stage between fusion-positive cases and controls. Notably, no other mutation driver was identified in P:R fusion-positive patients, showing mutual exclusiveness. However, one case of microsatellite instability-high (MSI-H) was identified in these P:R fusion positive patients, implying the possibility of the co-occurrence of two oncogenic aberrations.

**Table 1. Clinicopathological characteristics of *PTPRK-RSPO3* fusion-positive and fusion-negative cases in TCGA colorectal cancer.**

| | *Fusion (N = 7) | **Control (N = 186) | ***p values |
|---|---|---|---|
| Age | 46~76 | 31~90 | NS |
| Sex | | | 1 |
| • Male | 3/5 (60.0%) | 68/121 (56.2%) | |
| • Female | 2/5 (40.0%) | 53/121 (43.8%) | |
| Vital status | | | NS |
| • Alive | 5/5 (100.0%) | 101/121 (83.8%) | |
| • Dead | 0/5 (0.0%) | 20/121 (16.5%) | |
| Stage | | | NS |
| • Stage I | 0/4 (0.0%) | 20/116 (17.2%) | |
| • Stage II | 2/4 (50.0%) | 48/116 (41.4%) | |
| • Stage III | 2/4 (50.0%) | 33/116 (28.4%) | |
| • Stage IV | 0/4 (0.0%) | 15/116 (13.0%) | |
| Microsatellite instability | | | NS |
| • MSI-high | 1/5 (20.0%) | 18/121 (14.5%) | |
| • MSI-low | 0/5 (0.0%) | 19/121 (15.9%) | |
| • MSS | 4/5 (80.0%) | 84/121 (69.4%) | |
| Histological type | | | NS |
| • Adenocarcinoma | 3/4 (75.0%) | 113/120 (88.0%) | |
| • Mucinous adenocarcinoma | 1/4 (25.0%) | 7/120 (12.0%) | |
| Mutation profile | | | |
| • TP53 mutation | 3/7 | 113/186 | NS |
| | (42.9%) | (60.8%) | |
| • KRAS mutation | 2/7 | 80/186 | NS |
| | (28.6%) | (19.4%) | |
| • PIK3CA mutation | 2/7 | 50/186 | NS |
| | (28.6%) | (43.0%) | |
| • PTEN mutation | 1/7 | 15/186 | NS |
| | (14.3%) | (8.0%) | |
| • BRAF mutation | 2/7 | 23/186 | NS |
| | (28.6%) | (12.7%) | |

* Samples harboring PTPRK-RSPO3 fusion

** Control group was extracted from samples demonstrating the lower median of RSPO3 mRNA expressions.

***p-value was calculated between fusion positive samples and controls using moonBook R package.

Abbreviations: AD, adenocarcinoma; NS, not significant; NA, not available

## Key genes and pathways altered in PTPRK-RSPO3 fusion-positive colorectal cancer

By the correlations test and permutation tests (See Methods), 2,505 genes were passed the Pearson correlation-test with an R-value greater than the cut-off value. Eighteen genes including *PPP1R12B, NPY, VIP, C2orf72, IQGAP2, SYT2, ADCYAP1, ZNF385D, SCIN, MAGEE2, SDCBP2, AHCYL2, C6orf105, ZNF229, BTNL8, SLC7A14, GPR88,* and *ASTN1* showed good correlation (R > 0.5) with *RSPO3* in RNA expression (S1 Table).

In pathway analysis, ten different pathways were shown to be statistically significant: apoptosis-related pathway, direct p53-related pathway, EGFR-related pathway, ErbB-related pathway, JAK-STAT-related pathway, JAK-STAT-related pathway, tyrosine kinases-related

pathway, pathways in cancer, SCF-KIT-related pathway, VEGFR-related pathway, and WNT-related pathway. Of these pathways, the P:R fusion-positive cases in comparison to the P:R fusion-negative control demonstrated 848 significantly over- or under-expressed RNA expressions of 848 genes (Figs 1 and 2 and S1 and S2).

Among the 848 genes, 36 genes were annotated as cancer genes using the cancer gene census database offered by the CatalogueOfSomaticMutationsInCancer DB (COSMIC, https://cancer.sanger.ac.uk). Of these, ten genes from highest R-values were as follows: *ALK*, *ACSL3*, *AXIN2*, *PTPRK*, *CDX2*, *MYC*, *TP53*, *GNAQ*, *ACVR2A*, and *FAS*. RSPO3-correlated cancer genes involved in more than four pathways were as follows; *JUN* was involved the most in 9 of 10 pathways, *APC*, 6 pathways, *AXIN2*, 5 pathways, *FGFR2*, 5 pathways, *JAK2*, 7 pathways, *MDM2*, 5 pathways, *MYC*, 9 pathways, *RAC1*, 8 pathways, and lastly, *TP53*, 7 pathways. In addition, 4 genes were associated with 4 pathways, 4 genes in 3 pathways and 7 genes in 2 pathways (S3 Fig). Amongst these genes, those with a correlation R-value greater than 0.3 were *ALK*(R = 0.44), *ACSL3*(R = 0.43), *AXIN*(R = -0.38), *MYC* (R = -0.34), *TP53* (R = -0.33), *GNAQ* (R = 0.31), *ACVR2A* (R = 0.31), and *FAS* (R = 0.31).

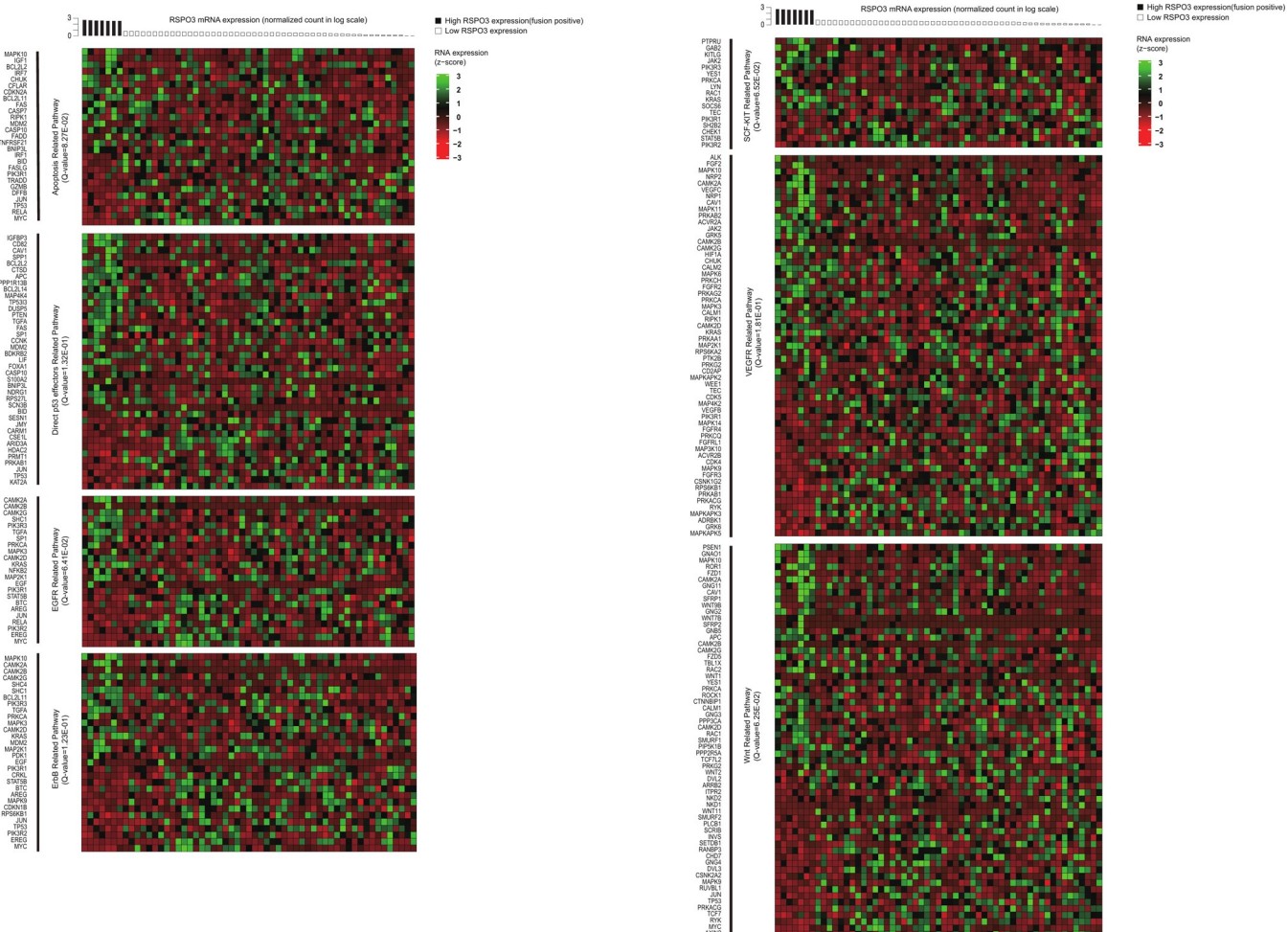

**Fig 1. Gene expression heatmap of 7 cancer-related pathways enriched with genes that were correlated to RSPO3 in RNA expression.** A total of 256 genes associated with Apoptosis, Direct p53, EGFR, ErbB, SCF-KIT, VEGFR, WNT signaling showed significant differences in expression between RSPO3 fusion-positive colorectal samples and the control samples (see details in Methods). The RNA expression was transformed to z-score. The x-axis represents the sample, and the y-axis represents the RNA expression.

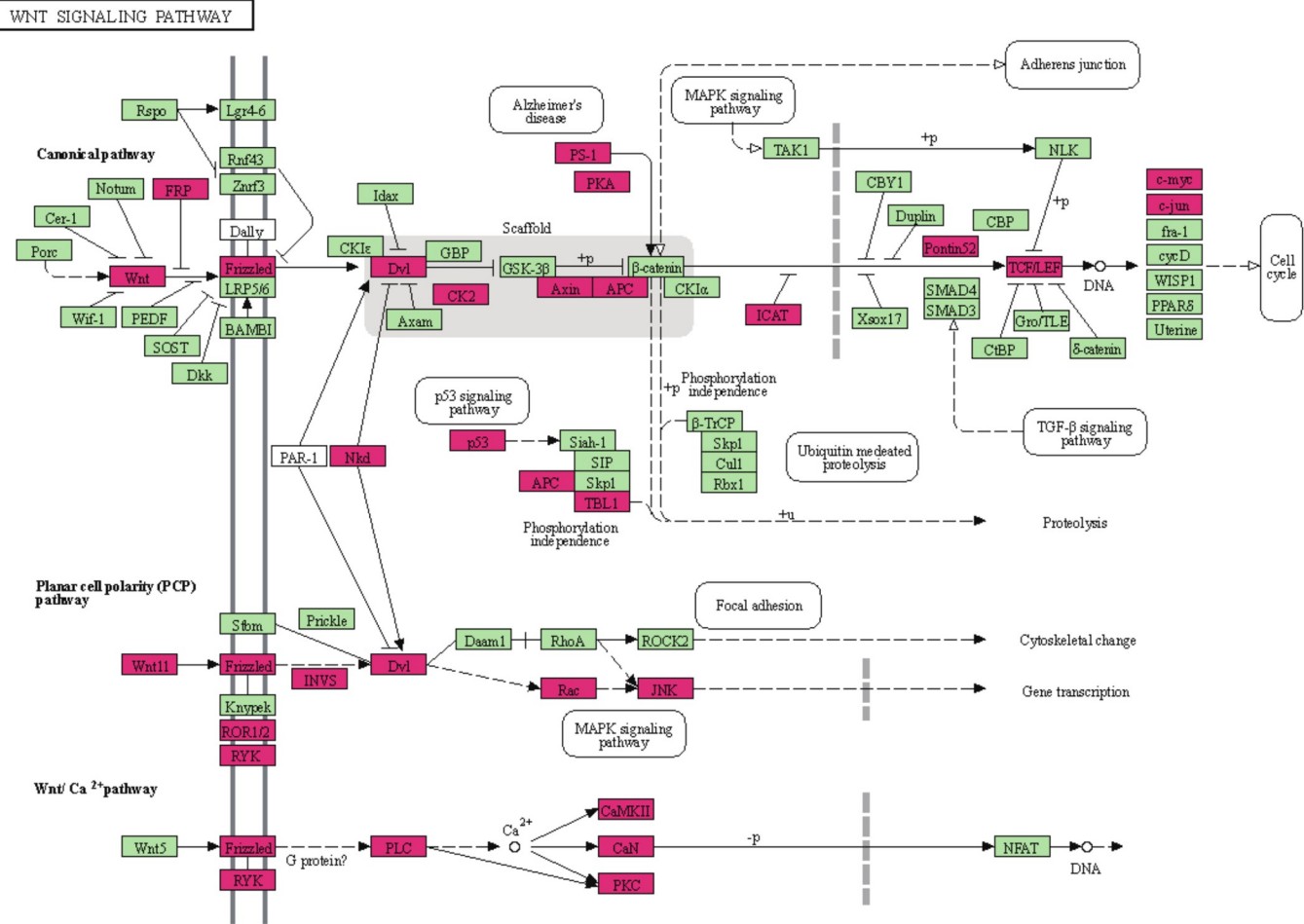

**Fig 2. Over- and under-expressed genes are highlighted in WNT signaling pathway.** The KEGG pathway map for the human WNT signaling pathway (hsa 04310) was illustrated using the KEGG Mapper; genes correlated with RSPO3 expression are colored in pink.

## Identification of therapeutic targets and inferring repurposed drug candidates

By matching the 848 genes included in the 10 pathways associated with P:R fusion-positive colorectal cancer using the CiVIC database and OncoKB, we were able to infer 673 and 262 drugs to have actionable target potential.

In the CIViC database, following 19 genes among 848 genes were related with actionable drugs: *ALK, FGFR2, TP53, HIF1A, EPAS1, KRAS, CEBPA, NOTCH1, STK11, JAK2, PGR, RAD50, PIK3R1, CDKN1B, NQO1, NT5E, MAP2K1, GNAQ,* and *PTEN. ALK* was identified to be a class A-type drug (Proven/consensus association in human medicine) which can be targeted using crizotinib, alectinib and ceritinib. In other class-type (B, C, D, and E) gene-drug association, additional thirteen drugs were found, considering the scenario for inhibitors for activated oncogenes or activators for down-regulated tumor suppressor genes (Fig 3A).

When using the OncoKB database, 4 genes (*KRAS, FGFR2, ALK,* and *JAK2*) were identified and they were all included in the inferred results using CIViC database. Level 1 drugs (FDA-recognized biomarker predictive of response to an FDA-approved drug) for target genes are as follows: lorlatinib, brigatinib, crizotinib, ceritinib, alectinib for *ALK*; erdafitinib, infigratinib,

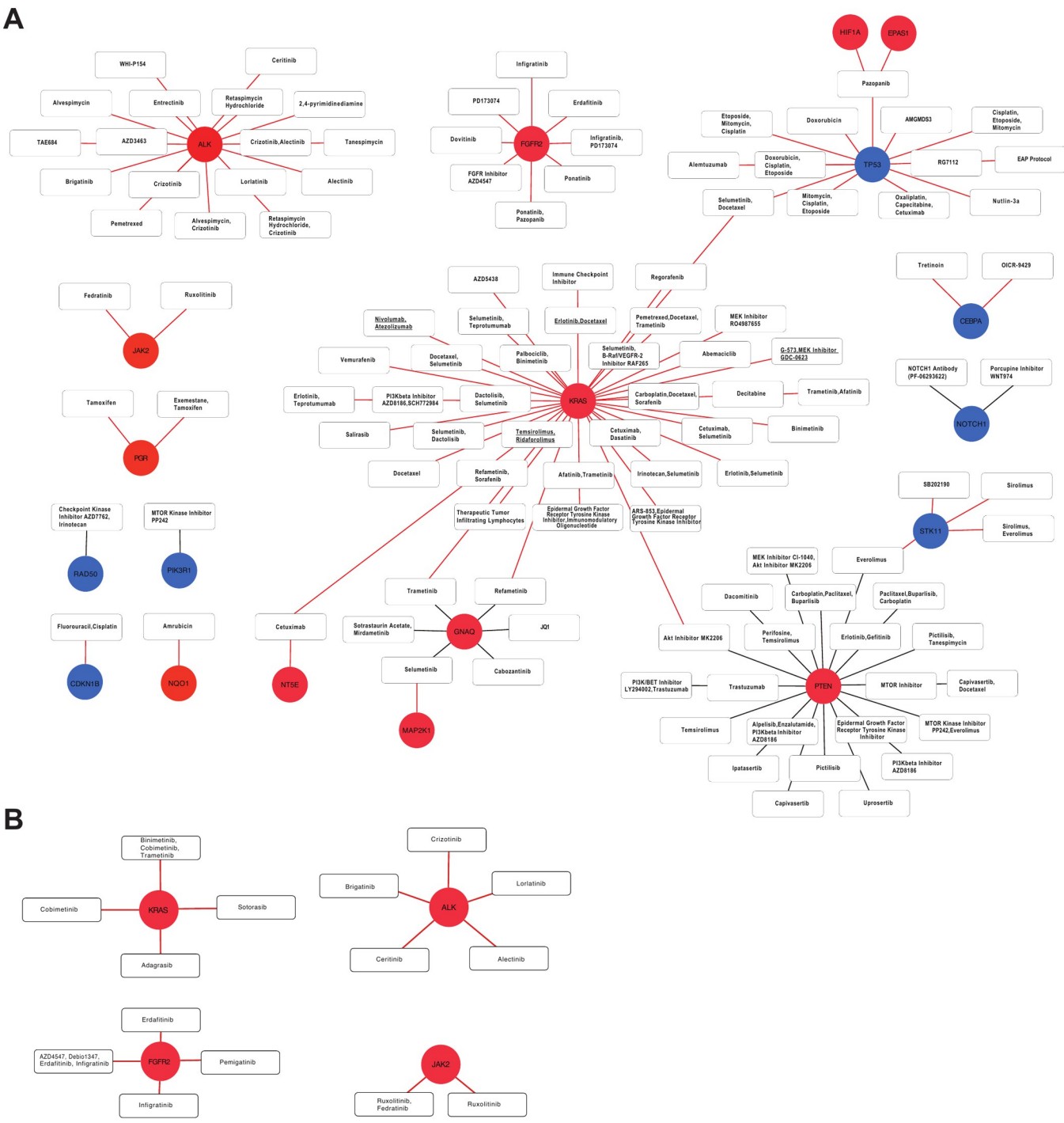

**Fig 3. Inferred drug-target network in PTPRK-RSPO3 fusion-positive colorectal cancer.** Drug-target relation was obtained based on CIViC and OncoKB databases: White boxes, drugs; circles, underlined white boxes, substitute drugs; genes; red circles, genes that are over-expressed in fusion-positive cancer; blue circles, genes that are under-expressed in fusion-positive cancer. The red lines are prioritized drug-target relationships based on the scenario that properly working cancer drugs are generally inhibitors for activated oncogenes or activators for down-regulated tumor suppressor genes.

pemigatinib for *FGFR2*; sotorasib for *KRAS*. These cancer drugs appeared to be inhibitors for activated oncogenes (Fig 3B).

Of 19 druggable genes, ten were involved in the multiple pathway: *PIK3R1* for 8 cancer-related pathways (Apoptosis, EGFR, ErbB related Pathway, JAK-STAT, Pathways in Cancer, SCF-KIT, Tyrosine kinases, VEGFR related pathway); *KRAS* for 6 cancer-related pathways (EGFR, ErbB related Pathway, Pathways in Cancer, SCF-KIT, Tyrosine kinases, VEGFR related pathway); *JAK2* for 5 cancer-related pathways (JAK-STAT pathway, Pathways in Cancer, SCF-KIT, Tyrosine kinases, VEGFR related pathway); *TP53* for 5 cancer-related pathways (Apoptosis, Direct p53, ErbB related Pathway, Pathways in Cancer, WNT-related Pathway); *MAP2K1* for 5 cancer-related pathways (EGFR, ErbB related Pathway, JAK-STAT, Pathways in Cancer, VEGFR related pathway); *FGFR2* for 3 cancer-related pathways (Tyrosine kinases, pathways In cancer, VEGFR related pathway); *ALK* for 2 cancer-related pathways (VEGFR related pathway and pathways in cancer); *HIF1A* for 2 cancer-related pathways (VEGFR related pathway and pathways in cancer); *CDKN1B* for 2 cancer-related pathways (ErbB related pathway and Pathways in Cancer); *PTEN* for 2 cancer-related pathways (Direct p53 and Pathways in Cancer).

## Discussion

In this study, drug candidates were identified in P:R fusion colorectal cancer as follows. First, genes correlated with *RSPO3* RNA expression were extracted, and oncogenic cell signaling pathways including these genes were selected. We then used the drug target database to build a drug target network in P:R fusion colorectal cancer, and prioritize suitable therapeutics (Fig 4). As a result, this study is expected to provide an opportunity to try a wider range of therapeutics in colorectal cancer, where EGFR inhibitors and ICI are limitedly used as targeted therapeutics [8].

Previous studies that systematically explore gene biomarkers with bioinformatics analysis in colorectal cancer have focused on discovering prognosis-related biomarkers using differentially expressed genes (DEGs) analysis and machine learning techniques [26, 27]. To our best knowledge, our study differs from previous studies in two respects. First, the purpose of this study is to discover novel targets and therapeutics related to original mutations by analyzing downstream pathways and genes affected by target mutations that cannot be directly targeted. Second, our study is based on a structural variation (P:R fusion by DNA structural variation) that is a driver mutation in colorectal cancer. As consequence, almost all genes correlated with P:R fusion are downstream-level genes affected by fusion. In this aspect, our study is different from other studies, and, for example, it is not clear whether COL11A1 is a primary driver or is affected by other drivers in the study by Ritwik et al [27].

The WNT signaling pathway is an important mediator in tissue homeostasis and recovery while it acts an important role in tumor-development of colorectal cancer [18]. Both in vitro experiments in human-colon cancer cell line HT-29 and in vivo experiments in CRISPR-based xenograft mice provided the evidence that *RSPO3* fusion gene was involved in the initiation and development of CRC via activating WNT signaling [14, 19]. This means that human CRC is a sensitive tumor for WNT-targeted treatment, suggesting that *RSPO3* fusion gene can be an effective therapeutic target.

As a result of our analysis, it is interesting that *ALK* up-regulated in P:R fusion-positive CRC has the following three characteristics. First, the correlation between *ALK* RNA expression and *RSPO3* RNA expression in P:R fusion-positive CRC was the highest among COSMIC common oncogenes (R = 0.44). Second, *ALK* was a gene involved in multiple cancer pathways. Finally, *ALK* inhibitors are FDA-approved therapeutics that perform well in other carcinomas (e.g. lung cancer) [28, 29]. Taken together, in-silico analysis showed that *ALK* inhibitors were highly likely to act in P:R fusion positivity [30].

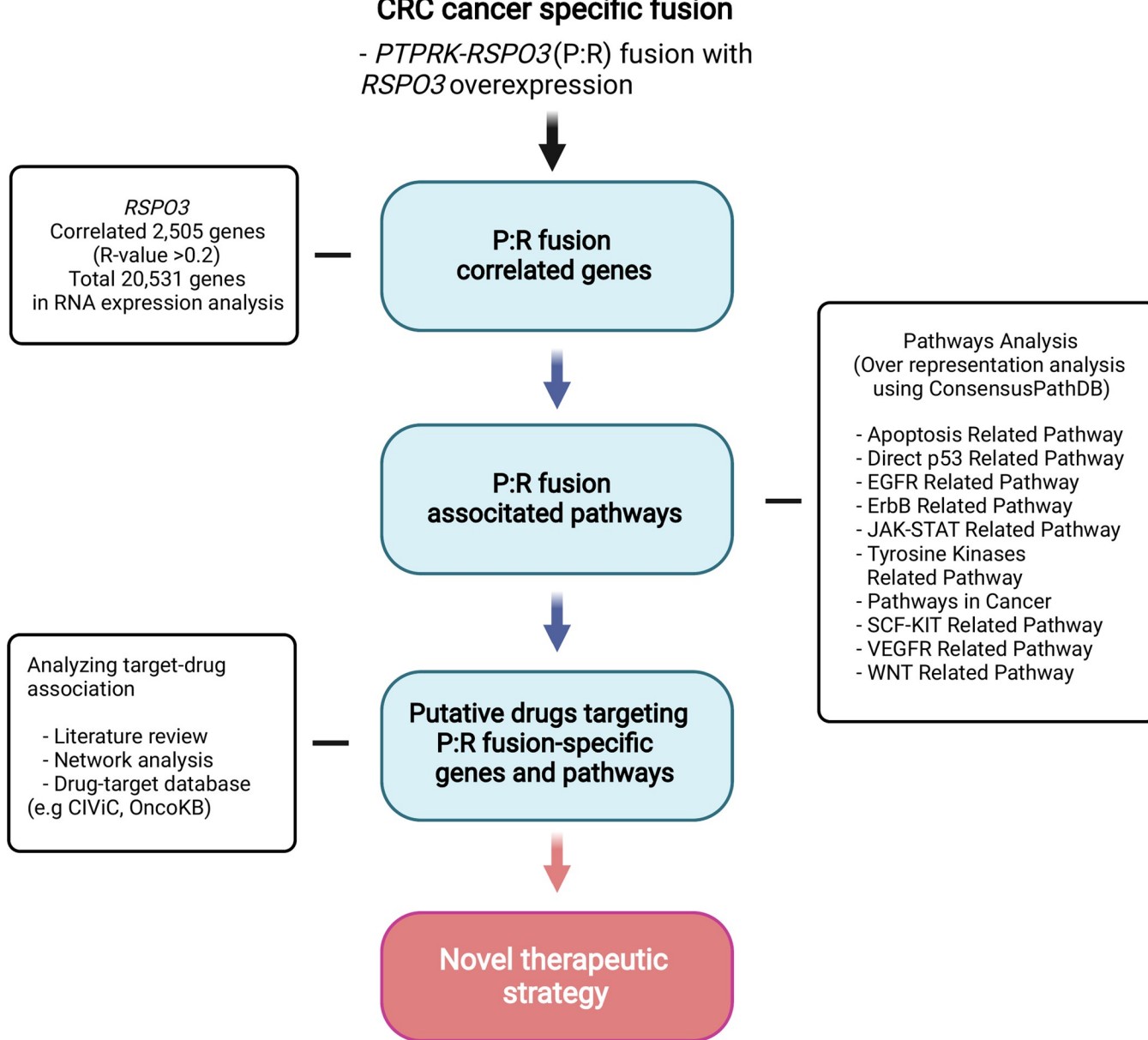

**Fig 4. Overall design of this study.** Transcriptome data for colorectal cancer (CRC) was attained from the Broad GDAC Firehose database. Following the RNA expression analysis of a total of 20,531 genes, 2,505 genes correlated with RSPO3 expression were selected. (R-value > 0.2, see Methods). Over-representation analysis of the 2,505 genes showed significant relation to 10 major cancer-related pathways (Apoptosis Related Pathway, Direct p53 Related Pathway, EGFR Related Pathway, ErbB Related Pathway, JAK-STAT Related Pathway, JAK-STAT Related Pathway, Tyrosine Kinases Related Pathway, Pathways in Cancer, SCF-KIT Related Pathway, VEGFR Related Pathway, WNT Related Pathway). Potential targets and repurposed drugs were inferred by analyzing target-drug associations via literature reviews and network analysis using the differentially expressed gene list and target-drug databases.

Despite the limited number of samples, the clinical characteristics of P:R-positive and P:R-negative patients were found to be similar. This indicates that even if the clinical properties are similar, the molecular properties may be different, which may require treatment to target the molecular properties. One interesting point is that P:R fusions can also be found in MSI-H. In this case, further clinical evaluation is needed to determine if there is a synergistic effect between ICI and the targeted therapy we propose.

In summary, we were able to present key indicators and clinically viable therapeutics for P:
R fusion-positive CRC. Our findings will serve as a steppingstone for future research in the
development of precision medicine targeting colorectal cancer.

## Supporting information

**S1 Fig. Gene expression heatmap of cancer-related pathways enriched with genes corre-
lated to RSPO3 in RNA expression.**
(PDF)

**S2 Fig. The KEGG pathway maps for the human ERBB signaling pathway and pathways in
cancer using the KEGG Mapper; genes correlated with RSPO3 expression are colored in
pink.**
(PDF)

**S3 Fig. Putative target genes involved in multiple pathways of PTPRK-RSPO3 fusion-posi-
tive cancer.**
(PDF)

**S4 Fig. Inferred drug-target network in PTPRK-RSPO3 fusion-positive colorectal cancer
based on VICC database.**
(PDF)

**S1 Table. 2,505 genes correlated with RSPO3 (R >0.2).**
(XLSX)

**S2 Table. Putative target genes and actionable drugs involved in ten major cancer-related
pathways.**
(XLSX)

## Author Contributions

**Conceptualization:** Jae Heon Jeong, Jae Won Yun, Chan Yeong Heo, Sejoon Lee.

**Data curation:** Jae Heon Jeong, Jae Won Yun, Sejoon Lee.

**Formal analysis:** Jae Won Yun.

**Investigation:** Jae Heon Jeong, Sejoon Lee.

**Methodology:** Jae Heon Jeong, Jae Won Yun, Chan Yeong Heo, Sejoon Lee.

**Project administration:** Chan Yeong Heo, Sejoon Lee.

**Resources:** Jae Heon Jeong, Sejoon Lee.

**Software:** Sejoon Lee.

**Supervision:** Jae Heon Jeong, Jae Won Yun, Sejoon Lee.

**Validation:** Ha Young Kim, Chan Yeong Heo, Sejoon Lee.

**Visualization:** Jae Heon Jeong, Jae Won Yun, Chan Yeong Heo, Sejoon Lee.

**Writing – original draft:** Jae Heon Jeong, Jae Won Yun, Sejoon Lee.

**Writing – review & editing:** Jae Heon Jeong, Jae Won Yun, Ha Young Kim, Chan Yeong Heo.

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
