## [Decision Letter · Decision Letter 0]

16 May 2022

PONE-D-22-06899Investigation of cell signalings and therapeutic targets in PTPRK-RSPO3 fusion-positive colorectal cancer.PLOS ONE

Dear Dr. Lee,

Thank you for submitting your manuscript to PLOS ONE. After careful consideration, we feel that it has merit but does not fully meet PLOS ONE’s publication criteria as it currently stands. Therefore, we invite you to submit a revised version of the manuscript that addresses the points raised during the review process.

The manuscript needs revision as suggested by the reviewer and myself. Please revise the manuscript  and submit. However, submitting a revision does not guarantee the acceptance of the same. 

We look forward to receiving your revised manuscript.

Kind regards,

Suprabhat Mukherjee, Ph.D.

Academic Editor

PLOS ONE

Journal Requirements:

“This study was supported by a VHS Medical Center Research Grant, Republic of Korea (VHSMC 21038), grant no 18-2018-023 from the SNUBH Research Fund, and the National Research Foundation of Korea(NRF) grant funded by the Korea government(MSIT)(No. 2022R1C1C1012986).”

“This study was supported by a VHS Medical Center Research Grant, Republic of Korea (VHSMC 21038), grant no 18-2018-023 from the SNUBH Research Fund, and the National Research Foundation of Korea(NRF) grant funded by the Korea government(MSIT)(No. 2022R1C1C1012986).”

“This study was supported by a VHS Medical Center Research Grant, Republic of Korea (VHSMC 21038), grant no 18-2018-023 from the SNUBH Research Fund, and the National Research Foundation of Korea(NRF) grant funded by the Korea government(MSIT)(No. 2022R1C1C1012986).”

5. One of the noted authors is a group or consortium “Chan Young Heo, Jae Won Yun”. In addition to naming the author group, please list the individual authors and affiliations within this group in the acknowledgments section of your manuscript. Please also indicate clearly a lead author for this group along with a contact email address.

Additional Editor Comments (if provided):

Oncogenesis events in CRC is indeed multifaceted and considering the impact of P:R fusion-positive CRC authors need to present a comparative view with the recently published papers citing the association of multiple signaling pathways in the course of CRC pathophysiology. Authors may follow and cite DOI: 10.3389/fgene.2021.608313 as well as relevant literature to improve the manuscript. Fig. 1 could be omitted and a scheme may be added at the last.

Reviewers' comments:

Reviewer's Responses to Questions

**Comments to the Author**

1. Is the manuscript technically sound, and do the data support the conclusions?

Reviewer #1: Partly

2. Has the statistical analysis been performed appropriately and rigorously? 

Reviewer #1: N/A

3. Have the authors made all data underlying the findings in their manuscript fully available?

Reviewer #1: Yes

4. Is the manuscript presented in an intelligible fashion and written in standard English?

Reviewer #1: Yes

5. Review Comments to the Author

Reviewer #1: Jeong HJ et al. presented key indicators and clinically viable therapeutics for P:R fusion-positive CRC as well as inferred the actionable drugs in bioinformatics analysis using the Cancer Genome Atlas (TCGA) data. Through drug-target network analysis, several putative therapeutic candidates were shown effective as they were applied and involved in multiple cancer-pathways.

The manuscript is generally well written and structured. However, in my opinion the data have some shortcomings in regards to some analyses, explanation, and logical sense.

Comments:

Line 82: the following sentences are not clear. 372 tumor samples with the barcode 01A were chosen, with other types of tumor samples, 11A (Normal) or 06A (Metastasized), being excluded. What do these numbers and barcodes referred to?

- How was the permutation test calculated?

As stated by the authors, the median of absolute R values, mostly affected 2,505 genes were selected by correlation cut-off (R > 0.2). What is the criteria to set a cut-off of R > 0.2?

- What does median in Table S1 imply?

Line 128: As stated, 7 patients demonstrated presence of the fusion mutation, whereas the remaining 416 patients were negative for the fusion based on the clinicpathological characteristics. These cases are not fully presented in the Table 1. The characteristics of PTPRK-RSPO3 fusion-positive and fusion-negative cases in TCGA colorectal cancer is not clear in Table 1. Moreover, what does median in Table S1 imply?

All the supplementary Tables and figures should be renamed as indicated in Sup section including tittle of each and applied methods for data extraction.

As Table 1 shows, none of the clinicopathological characteristics of selected cases is significant. Eighteen genes showed good correlation (R > 0.5) with RSPO3 in RNA expression. Is that correlation defines any significant RNA expression?

Line 153: How come the ten different pathways were shown to be statistically significant? How was the significant means elaborated?

Line 163: the following sentence “ten genes from highest R-values were as follows” is not clear. What is the implication of highest R-values?

Fig 4 represents Drug-target relation obtained based on Civic and OncoKB databases. What was the intention to select these two databases? The VICC meta-database has recently been reported the most extensive source of information, featuring 92% of variants with a drug association (https://dx.doi.org/10.21873%2Fcgp.20250). Is there any explanation?

Line 184: 4 genes (KRAS, FGFR2, ALK, and JAK2) were identified and they were all included in the inferred results using CIViC database. How the authors link the gene presence with their RNA expression involved in several pathways as well as disease conditions?

Line 191: Of 19 druggable genes, five were involved in the multiple pathways. Have these genes as well as their related pathways experimentally reported?

6. PLOS authors have the option to publish the peer review history of their article (what does this mean?). If published, this will include your full peer review and any attached files.

Reviewer #1: **Yes: **Meysam Sarshar

---

## [Author Response · Author response to Decision Letter 0]

29 Jun 2022

Response to Reviewer Comments

 We deeply appreciate the reviewer Meysam Sarshar’s attention to our paper. Below, we address each of the reviewer's comments point by point. We revised our manuscript using MS Word and highlighted the changes with yellow color.

Point 1: Line 82: the following sentences are not clear. 372 tumor samples with the barcode 01A were chosen, with other types of tumor samples, 11A (Normal) or 06A (Metastasized), being excluded. What do these numbers and barcodes referred to? 

Response 1: Thank you for your comment. For better understanding of the TCGA data for readers, the following sentence has been modified as follows (Line: 78-79):

“Of the 433 GDAC data downloaded, 53 normal and 1 metastasis data were omitted, and the remaining 379 data were analyzed.”

The barcode contains information concerning the institution, patient, sample, and method of analysis. For instance, the two digits from 01A, 06A, 11A barcodes reveal information on the type of patient sample whether it be normal or tumor, and the alphabet indicates the vial containing the patient sample (https://gdc.cancer.gov/resources-tcga-users/tcga-code-tables/sample-type-codes). Although the majority of the TCGA data were tumor samples including 01-09, a portion of them, 10-19 were normal samples. Therefore, it was necessary to select and assess samples according to the purpose of analysis. We extracted primary solid tumor samples, which have barcodes 01, used for this study. 

Point 2: How was the permutation test calculated? As stated by the authors, the median of absolute R values, mostly affected 2,505 genes were selected by correlation cut-off (R > 0.2). What is the criteria to set a cut-off of R > 0.2?

Response 2: Thank you for your detailed comment that improving our study. Of the 57 samples, we selected two genes and their RNA expression values were randomly ordered respectively, and Pearson correlation coefficient R value was obtained between two genes. This process was repeated 100,000 times. The resulting frequency of an R value higher than 0.2 by chance was 378 out of 100,000 times. Among the 2,505 genes that showed a correlation with RSPO3 of an R value of 0.2 or higher, approximately 0.378%, that is 20,531 gene multiplied by 0.00378 were estimated to yield 78 false positives. Hence, of the 2,505 genes, 2,427 genes may have a correlation with RSPO3. 

Point 3: What does median in Table S1 imply? (Moreover, what does median in Table S1 imply?)

Response 3: Thank you for your comment. For instance, should KRAS mutation-positive samples occupy a majority of the 50 samples that were selected as control, there is a high likelihood for a large number of KRAS signature to be generated. In order to minimize the bias from undesired signature accumulation, 50 samples from the 186 samples expressing low RSPO3 expression were randomly selected as the control. Pearson R values were calculated between reference gene (RSPO3) and the other genes. Above process was repeated 100 times and the median R values were calculated in each gene. 

Point 4: Line 128: As stated, 7 patients demonstrated presence of the fusion mutation, whereas the remaining 416 patients were negative for the fusion based on the clinicopathological characteristics. These cases are not fully presented in the Table 1. The characteristics of PTPRK-RSPO3 fusion-positive and fusion-negative cases in TCGA colorectal cancer is not clear in Table 1.

Response 4: Thank you for valuable comment. According to your comment, table 1 has been revised (Line: 138-140). The number of fusion and control samples were more clearly described in our study. The TCGA colorectal RNA samples used were extracted from the GDAC firehose. Of them, 379 were tumor solid samples, and of the 372 samples, excluding the 7 fusion-positive samples, 186 samples expressing less than 50% RSPO3 mRNA expression were selected as control (fusion-positive). The clinicopathological analysis was performed according to the criteria above and reflected in Table 1 and its pertinent text.

We modified the sentences on line 123-126 as follows:

“The clinicopathological characteristics in this study were described in Table 1. A total 379 tumor solid samples, and of the 372 samples, excluding the 7 fusion-positive samples, 186 samples expressing less than 50% RSPO3 mRNA expression were selected as control.” 

Point 5: All the supplementary Tables and figures should be renamed as indicated in Sup section including tittle of each and applied methods for data extraction.

Response 5: Thank you for your comment. We revised the supplementary section on line 263-275 as follows:

S1 Fig: Gene expression heatmap of cancer-related pathways enriched with genes correlated to RSPO3 in RNA expression. 

S2 Fig: The KEGG pathway maps for the human ERBB signaling pathway and pathways in cancer using the KEGG Mapper; genes correlated with RSPO3 expression are colored in pink. 

S3 Fig: Putative target genes involved in multiple pathways of PTPRK-RSPO3 fusion-positive cancer.

S4 Fig: Inferred drug-target network in PTPRK-RSPO3 fusion-positive colorectal cancer based on VICC database. 

S1 Table: 2,505 genes correlated with RSPO3 (R >0.2). 

S2 Table: Putative target genes and actionable drugs involved in ten major cancer-related pathways.

Point 6: As Table 1 shows, none of the clinicopathological characteristics of selected cases is significant. Eighteen genes showed good correlation (R > 0.5) with RSPO3 in RNA expression. Is that correlation defines any significant RNA expression?

Response 6: Thank you for your comment. Since an overexpression of RSPO3 mRNA level is observed in PTPRK-RSPO3 fusion samples (PMID: 32631391 Figure 2B), we assumed that downstream genes affected by upregulated RSPO3 expression with fusion would have high R values in expression level by the Pearson correlation test. Due to the limited number (n=7) of RSPO3 fusion samples, it was not ideal to infer a statistically significant difference from clinical variable. 

Point 7: Line 153: How come the ten different pathways were shown to be statistically significant? How was the significant means elaborated?

Response 7: Thank you for your comment. In summary, a P-value is calculated using the hypergeometric distribution. The P-value reflects the significance of the observed overlap between the input gene list and the module's members when compared to random expectations. A low P-value indicates that more of the module's members are present in the input list than would be expected by chance. The p-values are corrected for multiple testing using the false discovery rate method.

Detailed explanation: PMID: 18940869

Point 8: Line 163: the following sentence “ten genes from highest R-values were as follows” is not clear. What is the implication of highest R-values?

Response 8: Thank you for your comment. Since mRNA overexpression of RSPO3 occurs by PTPRK-RSPO3 fusion (PMID: 32631391 Figure 2B), it was determined that genes affected by expression downstream by upregulated RSPO3 would have high R values in the Pearson correlation test. The reason for selecting the 10 highest R-value genes is to find the most likely downstream pathway or target in the absence of a direct target therapy for RSPO3. Genes with the highest R values were extracted with the purpose of identifying therapeutic targets from the downstream pathway of RSPO3.

Point 9: Fig 4 represents Drug-target relation obtained based on Civic and OncoKB databases. What was the intention to select these two databases? The VICC meta-database has recently been reported the most extensive source of information, featuring 92% of variants with a drug association (https://dx.doi.org/10.21873%2Fcgp.20250). Is there any explanation?

Response 9: Thank you for your comment. The data provided by VICC meta-database is in the form of a JSON file. However, since the JSON files were broken and not unified in ordinary format, we had to manually parse every five databases (jax, brca, cgi, molecularmatch, pmkb) except CIViC and OncoKB. Then, we were only able to extract information about genes and drugs from jax, brca and molecularmatch databases. However, advanced information such as clinical significance, evidence direction and evidence type were limited to parse. For this reason, the analyzed results using VICC database were not included in the main figure but were included in the supplementary figures. We added explanation about VICC figures in our manuscript (Line: 270, 295).

Point 10: Line 184: 4 genes (KRAS, FGFR2, ALK, and JAK2) were identified and they were all included in the inferred results using CIViC database. How the authors link the gene presence with their RNA expression involved in several pathways as well as disease conditions?

Response 10: Thank you for your comment. Generally, there are three classes for activation signaling including hotspot mutation, amplification, and overexpression. For this study, RNA sequence-based overexpression was considered an activating signal. The drug database provides information on the relationship between the activating signaling (three classes mentioned above) and available drugs. For example, MET activating mutations are including amplification, over-expression, and activating point mutations and the three class of mutations are mostly sharing target-drug sensitivity (capmatinib, tepotinib). So, the three types of mutations were considered as showing similar target-drug sensitivity in silico level. Although the sensitivity of the drugs may differ according to the various types of signal activation, the purpose of this study is to enroll as many drugs with high potential as possible. It would be ideal for these hypotheses to be validated with further additional experimentations. However, the scope of this study does not encompass validation experiments and will take into consideration for future studies.

Point 11: Line 191: Of 19 druggable genes, five were involved in the multiple pathways. Have these genes as well as their related pathways experimentally reported?

Response 11: Thank you for your comment. While reviewing the content, five additional druggable genes involved multiple pathways were found and added to the manuscript (Line:217-230). The Consensus Path Database (CPDB, http://cpdb.molgen.mpg.de/CPDB) we utilized is a database combined by integrating 177 biological pathways based on experimentally reported studies: INOH (PMID: 2212066), KEGG (PMID: 27899662), NetPath (PMID: 20067622), PID (PMID: 18832364), Reactome (PMID: 26656494), and Wikipathways (PMID: 26481357). We attached a table of CPDB sources and 10 druggable genes involved in multiple pathways on below:

Gene Source Pathway

PIK3R1 KEGG Apoptosis

PIK3R1 KEGG VEGFR Related Pathway

PIK3R1 KEGG ErbB Related Pathway

PIK3R1 KEGG JAK-STAT Pathway

PIK3R1 KEGG EGFR

PIK3R1 KEGG Tyrosine Kinases

PIK3R1 KEGG Pathways In Cancer

PIK3R1 KEGG SCF-KIT

PIK3R1 Reactome Apoptosis

PIK3R1 Reactome VEGFR Related Pathway

PIK3R1 Reactome ErbB Related Pathway

PIK3R1 Reactome JAK-STAT Pathway

PIK3R1 Reactome EGFR

PIK3R1 Reactome Tyrosine Kinases

PIK3R1 Reactome Pathways In Cancer

PIK3R1 Reactome SCF-KIT

PIK3R1 Wikipathways Apoptosis

PIK3R1 Wikipathways VEGFR Related Pathway

PIK3R1 Wikipathways ErbB Related Pathway

PIK3R1 Wikipathways JAK-STAT Pathway

PIK3R1 Wikipathways EGFR

PIK3R1 Wikipathways Tyrosine Kinases

PIK3R1 Wikipathways Pathways In Cancer

PIK3R1 Wikipathways SCF-KIT

PIK3R1 PharmGKB Apoptosis

PIK3R1 PharmGKB VEGFR Related Pathway

PIK3R1 PharmGKB ErbB Related Pathway

PIK3R1 PharmGKB JAK-STAT Pathway

PIK3R1 PharmGKB EGFR

PIK3R1 PharmGKB Tyrosine Kinases

PIK3R1 PharmGKB Pathways In Cancer

PIK3R1 PharmGKB SCF-KIT

PIK3R1 PID Apoptosis

PIK3R1 PID VEGFR Related Pathway

PIK3R1 PID ErbB Related Pathway

PIK3R1 PID JAK-STAT Pathway

PIK3R1 PID EGFR

PIK3R1 PID Tyrosine Kinases

PIK3R1 PID Pathways In Cancer

PIK3R1 PID SCF-KIT

PIK3R1 INOH Apoptosis

PIK3R1 INOH VEGFR Related Pathway

PIK3R1 INOH ErbB Related Pathway

PIK3R1 INOH JAK-STAT Pathway

PIK3R1 INOH EGFR

PIK3R1 INOH Tyrosine Kinases

PIK3R1 INOH Pathways In Cancer

PIK3R1 INOH SCF-KIT

KRAS Reactome Pathways In Cancer

KRAS Reactome Tyrosine Kinases

KRAS Reactome ErbB Related Pathway

KRAS Reactome SCF-KIT

KRAS Reactome VEGFR Related Pathway

KRAS Reactome EGFR

KRAS KEGG Pathways In Cancer

KRAS KEGG Tyrosine Kinases

KRAS KEGG ErbB Related Pathway

KRAS KEGG SCF-KIT

KRAS KEGG VEGFR Related Pathway

KRAS KEGG EGFR

KRAS Wikipathways Pathways In Cancer

KRAS Wikipathways Tyrosine Kinases

KRAS Wikipathways ErbB Related Pathway

KRAS Wikipathways SCF-KIT

KRAS Wikipathways VEGFR Related Pathway

KRAS Wikipathways EGFR

KRAS PharmGKB Pathways In Cancer

KRAS PharmGKB Tyrosine Kinases

KRAS PharmGKB ErbB Related Pathway

KRAS PharmGKB SCF-KIT

KRAS PharmGKB VEGFR Related Pathway

KRAS PharmGKB EGFR

JAK2 KEGG SCF-KIT

JAK2 KEGG Tyrosine Kinases

JAK2 KEGG Pathways In Cancer

JAK2 KEGG VEGFR Related Pathway

JAK2 KEGG JAK-STAT Pathway

JAK2 Reactome SCF-KIT

JAK2 Reactome Tyrosine Kinases

JAK2 Reactome Pathways In Cancer

JAK2 Reactome VEGFR Related Pathway

JAK2 Reactome JAK-STAT Pathway

JAK2 INOH SCF-KIT

JAK2 INOH Tyrosine Kinases

JAK2 INOH Pathways In Cancer

JAK2 INOH VEGFR Related Pathway

JAK2 INOH JAK-STAT Pathway

JAK2 Wikipathways SCF-KIT

JAK2 Wikipathways Tyrosine Kinases

JAK2 Wikipathways Pathways In Cancer

JAK2 Wikipathways VEGFR Related Pathway

JAK2 Wikipathways JAK-STAT Pathway

TP53 KEGG Direct p53 effectors

TP53 KEGG ErbB Related Pathway

TP53 KEGG Wnt Related Pathway

TP53 KEGG Apoptosis

TP53 KEGG Pathways In Cancer

TP53 Wikipathways Direct p53 effectors

TP53 Wikipathways ErbB Related Pathway

TP53 Wikipathways Wnt Related Pathway

TP53 Wikipathways Apoptosis

TP53 Wikipathways Pathways In Cancer

TP53 PID Direct p53 effectors

TP53 PID ErbB Related Pathway

TP53 PID Wnt Related Pathway

TP53 PID Apoptosis

TP53 PID Pathways In Cancer

MAP2K1 Wikipathways JAK-STAT Pathway

MAP2K1 Wikipathways VEGFR Related Pathway

MAP2K1 Wikipathways Pathways In Cancer

MAP2K1 Wikipathways EGFR

MAP2K1 Wikipathways ErbB Related Pathway

MAP2K1 KEGG JAK-STAT Pathway

MAP2K1 KEGG VEGFR Related Pathway

MAP2K1 KEGG Pathways In Cancer

MAP2K1 KEGG EGFR

MAP2K1 KEGG ErbB Related Pathway

MAP2K1 PharmGKB JAK-STAT Pathway

MAP2K1 PharmGKB VEGFR Related Pathway

MAP2K1 PharmGKB Pathways In Cancer

MAP2K1 PharmGKB EGFR

MAP2K1 PharmGKB ErbB Related Pathway

MAP2K1 INOH JAK-STAT Pathway

MAP2K1 INOH VEGFR Related Pathway

MAP2K1 INOH Pathways In Cancer

MAP2K1 INOH EGFR

MAP2K1 INOH ErbB Related Pathway

FGFR2 KEGG Pathways In Cancer

FGFR2 KEGG VEGFR Related Pathway

FGFR2 KEGG Tyrosine Kinases

FGFR2 Reactome Pathways In Cancer

FGFR2 Reactome VEGFR Related Pathway

FGFR2 Reactome Tyrosine Kinases

FGFR2 PharmGKB Pathways In Cancer

FGFR2 PharmGKB VEGFR Related Pathway

FGFR2 PharmGKB Tyrosine Kinases

FGFR2 INOH Pathways In Cancer

FGFR2 INOH VEGFR Related Pathway

FGFR2 INOH Tyrosine Kinases

ALK KEGG VEGFR Related Pathway

ALK KEGG Pathways In Cancer

ALK INOH VEGFR Related Pathway

ALK INOH Pathways In Cancer

HIF1A PID VEGFR Related Pathway

HIF1A PID Pathways In Cancer

HIF1A KEGG VEGFR Related Pathway

HIF1A KEGG Pathways In Cancer

CDKN1B KEGG Pathways In Cancer

CDKN1B KEGG ErbB Related Pathway

CDKN1B Wikipathways Pathways In Cancer

CDKN1B Wikipathways ErbB Related Pathway

PTEN KEGG Direct p53 effectors

PTEN KEGG Pathways In Cancer

PTEN PID Direct p53 effectors

PTEN PID Pathways In Cancer

PTEN Wikipathways Direct p53 effectors

PTEN Wikipathways Pathways In Cancer

---

## [Editor Report · Decision Letter 1]

4 Jul 2022

PONE-D-22-06899R1Investigation of cell signalings and therapeutic targets in PTPRK-RSPO3 fusion-positive colorectal cancer.PLOS ONE

Dear Dr. Lee,

Thank you for submitting your manuscript to PLOS ONE. After careful consideration, we feel that it has merit but does not fully meet PLOS ONE’s publication criteria as it currently stands. Therefore, we invite you to submit a revised version of the manuscript that addresses the points raised during the review process.

We look forward to receiving your revised manuscript.

Kind regards,

Suprabhat Mukherjee, Ph.D.

Academic Editor

PLOS ONE

Journal Requirements:

Additional Editor Comments (if provided):

Authors need to address the following comments.

Oncogenesis events in CRC is indeed multifaceted and considering the impact of P:R fusion-positive CRC authors need to present a comparative view with the recently published papers citing the association of multiple signaling pathways in the course of CRC pathophysiology. Authors may follow and cite DOI: 10.3389/fgene.2021.608313 as well as relevant literature to improve the manuscript. Fig. 1 could be omitted and a scheme may be added at the last.
---

## [Author Response · Author response to Decision Letter 1]

18 Jul 2022

July 18th, 2022

Editor, Plos One

Dear Editor

We deeply appreciate your attention to our paper. Below, we address each of the reviewer's comments point by point. We revised our manuscript using MS Word and highlighted the changes with yellow color.

Point 1: Please review your reference list to ensure that it is complete and correct. If you have cited papers that have been retracted, please include the rationale for doing so in the manuscript text or remove these references and replace them with relevant current references. Any changes to the reference list should be mentioned in the rebuttal letter that accompanies your revised manuscript. If you need to cite a retracted article, indicate the article’s retracted status in the References list and also include a citation and full reference for the retraction notice.

Response 1: Thank you for your comment. We corrected a miscited 20th reference from “Wilson RG, Smith AN, Bird CC. Immunohistochemical detection of abnormal cell proliferation in colonic mucosa of subjects with polyps. J Clin Pathol. 1990;43(9):744-747. doi:10.1136/jcp.43.9.744” to “Yamamoto S, Sakai N, Nakamura H, Fukagawa H, Fukuda K, Takagi T. INOH: ontology-based highly structured database of signal transduction pathways. Database (Oxford). 2011;2011: bar052. Published 2011 Nov 26. doi:10.1093/database/bar052”. We also checked and confirmed other references.

Point 2: Oncogenesis events in CRC is indeed multifaceted and considering the impact of P:R fusion-positive CRC authors need to present a comparative view with the recently published papers citing the association of multiple signaling pathways in the course of CRC pathophysiology. Authors may follow and cite DOI: 10.3389/fgene.2021.608313 as well as relevant literature to improve the manuscript. 

Response 2: We appreciate your valuable comment. We reviewed the recent articles regarding integrated bioinformatics approach of colorectal cancer and presented a comparative view in the discussion section on line 241-252 as follows:

“Previous studies that systematically explore gene biomarkers with bioinformatics analysis in colorectal cancer have focused on discovering prognosis-related biomarkers using differentially expressed genes (DEGs) analysis and machine learning techniques. (26, 27). To our best knowledge, our study differs from previous studies in two respects. First, the purpose of this study is to discover novel targets and therapeutics related to original mutations by analyzing downstream pathways and genes affected by target mutations that cannot be directly targeted. Second, our study is based on a structural variation (P:R fusion by DNA structural variation) that is a driver mutation in colorectal cancer. As consequence, almost all genes correlated with P:R fusion are downstream-level genes affected by fusion. In this aspect, our study is different from other studies, and, for example, it is not clear whether COL11A1 is a primary driver or is affected by other drivers in the study by Ritwik et al (27).”

Point 3: Fig. 1 could be omitted, and a scheme may be added at the last.

Response 3: Thank you for your comment. We omitted Fig.1 and generated Fig.4 regarding a scheme of this study on line 228-238.

Additionally, we removed all the funding-related information in our manuscript and

would like to change the funding information in “Financial Disclosure” section as

follows:

“This study was supported by a VHS Medical Center Research Grant, Republic of

Korea (VHSMC22057), grant no 18-2018-023 from the SNUBH Research Fund, and The

National Research Foundation of Korea (NRF) grant funded by the Korea government

(MSIT) (No. 2022R1C1C1012986).”

Thank you again for your thoughtful consideration.

Best regards,

Sejoon Lee

Department of Pathology and Translational Medicine

Clinical Precision Medicine Center

Seoul National University Bundang Hospital

Tel:+82-31-787-8124

---

## [Editor Report · Decision Letter 2]

31 Aug 2022

Investigation of cell signalings and therapeutic targets in PTPRK-RSPO3 fusion-positive colorectal cancer.

PONE-D-22-06899R2

Dear Dr. Lee,

We’re pleased to inform you that your manuscript has been judged scientifically suitable for publication and will be formally accepted for publication once it meets all outstanding technical requirements.

Kind regards,

Suprabhat Mukherjee, Ph.D.

Academic Editor

PLOS ONE
---

## [Editor Report · Acceptance letter]

8 Sep 2022

PONE-D-22-06899R2 

Investigation of cell signalings and therapeutic targets in PTPRK-RSPO3 fusion-positive colorectal cancer. 

Dear Dr. Lee:

I'm pleased to inform you that your manuscript has been deemed suitable for publication in PLOS ONE. Congratulations! Your manuscript is now with our production department. 

Kind regards, 

on behalf of

Dr. Suprabhat Mukherjee 

Academic Editor

PLOS ONE